# HIF drives lipid deposition and cancer in ccRCC via repression of fatty acid metabolism

Weinan Du[1], Luchang Zhang[1], Adina Brett-Morris[1], Brittany Aguila[1], Janos Kerner[2], Charles L. Hoppel[2,3], Michelle Puchowicz[4], Dolors Serra[5,6], Laura Herrero [5,6], Brian I. Rini[7], Steven Campbell[8] & Scott M. Welford[1]

Clear cell renal cell carcinoma (ccRCC) is histologically defined by its lipid and glycogen-rich cytoplasmic deposits. Alterations in the VHL tumor suppressor stabilizing the hypoxia-inducible factors (HIFs) are the most prevalent molecular features of clear cell tumors. The significance of lipid deposition remains undefined. We describe the mechanism of lipid deposition in ccRCC by identifying the rate-limiting component of mitochondrial fatty acid transport, carnitine palmitoyltransferase 1A (CPT1A), as a direct HIF target gene. CPT1A is repressed by HIF1 and HIF2, reducing fatty acid transport into the mitochondria, and forcing fatty acids to lipid droplets for storage. Droplet formation occurs independent of lipid source, but only when CPT1A is repressed. Functionally, repression of CPT1A is critical for tumor formation, as elevated CPT1A expression limits tumor growth. In human tumors, CPT1A expression and activity are decreased versus normal kidney; and poor patient outcome associates with lower expression of CPT1A in tumors in TCGA. Together, our studies identify HIF control of fatty acid metabolism as essential for ccRCC tumorigenesis.

[1] Department of Radiation Oncology, Case Western Reserve University School of Medicine, 10900 Euclid Avenue, Cleveland, OH 44106, USA. [2] Department of Pharmacology, Case Western Reserve University School of Medicine, 10900 Euclid Avenue, Cleveland, OH 44106, USA. [3] Department of Medicine, Case Western Reserve University School of Medicine, 10900 Euclid Avenue, Cleveland, OH 44106, USA. [4] Department of Nutrition, Case Western Reserve University School of Medicine, 10900 Euclid Avenue, Cleveland, OH 44106, USA. [5] Department of Biochemistry and Physiology, Institut de Biomedicina de la Universitat de Barcelona (IBUB), Universitat de Barcelona, E-08028 Barcelona, Spain. [6] Centro de Investigación Biomédica en Red de Fisiopatología de la Obesidad y la Nutrición (CIBEROBN), Instituto de Salud Carlos III, E-28029 Madrid, Spain. [7] Department of Hematology and Oncology, Cleveland Clinic Foundation, 9500 Euclid Avenue, Cleveland, OH 44106, USA. [8] Department of Urology, Cleveland Clinic Foundation, 9500 Euclid Avenue, Cleveland, OH 44106, USA. Correspondence and requests for materials should be addressed to S.M.W. (email: scott.welford@case.edu)

Clear cell renal cell carcinoma (ccRCC) is the most common form of renal cancer, and the most deleterious tumor afflicting cancer-prone von Hippel–Lindau patients. Clear cell tumors are defined histologically as malignant epithelial cells with clear cytoplasm, owing to a vast accumulation of lipids and glycogen that are removed in standard histological preparations[1]. While great strides have been made in identifying the genetic alterations driving ccRCC development[2], the significance of, and molecular mechanisms leading to, the clear cell phenotype are incompletely appreciated.

The canonical molecular alteration in ccRCC is inactivation of the von Hippel–Lindau tumor suppressor (VHL) located on chromosome 3p. Whether due to genetic predisposition, as in the case of von Hippel–Lindau disease, or due to somatic mutations or methylation, VHL alterations have been estimated to occur in near 90% of all clear cell tumors[3, 4]. A principal role of VHL is in the regulation of hypoxia-inducible factors involved in oxygen sensing. As an E3 ubiquitin ligase, VHL inactivation leads to constitutive activation of HIF1 and HIF2 through the stabilization of oxygen labile HIFα subunits[5]. Subsequent activation of hypoxic gene expression downstream of HIF1 and HIF2 is thought to be a major driving force in ccRCC development, and has led to targeted therapeutic strategies aimed at the well-described HIF target gene vascular endothelial growth factor (e.g., Sunitinib) that have become the standard of care[6].

Gene expression programs activated by HIFs in cancer include angiogenesis, anaerobic metabolism, inflammation, and metastasis[7]. A recent analysis of programs altered in ccRCC compared to normal kidneys identified an adipogenic gene signature, and

led to studies that demonstrated that ccRCC cells can undergo trans-differentiation when exposed to established adipogenic differentiation protocols[8], suggesting some mechanistic insight into the lipid deposition phenotype. Notably, adipogenic differentiation in vitro is associated with terminal cessation of the cell cycle, unlike the behavior of tumor cells. Nonetheless, ccRCC clearly display a propensity for lipid deposition rather than lipid catabolism.

Fatty acid (FA) synthesis is an anabolic process that responds to excess citrate in the cytoplasm[9]. Metabolism of glucose under aerobic conditions produces pyruvate, which enters the citric acid cycle in mitochondria by the action of pyruvate dehydrogenase to produce acetyl-CoA, and then citrate-by-citrate synthase. Citrate can also be produced from metabolism of glutamine via α-ketoglutarate in cancer cells, either through forward flux through oxaloacetate, or through reverse cycle activity of isocitrate dehydrogenase and aconitase[10], as recently observed in ccRCC[11]. Excess mitochondrial citrate is exported to the cytosol, where it is a substrate for ATP citrate lyase to produce cytosolic acetyl-CoA. Subsequently carboxylation of acetyl-CoA by acetyl-CoA carboxylase to form malonyl-CoA is the commitment step in FA synthesis. The principal roles of FAs are to serve as substrates for membrane synthesis, energy stores, and production of signaling molecules. Abnormal cancer metabolism leads to changes in decisions regarding FA fates, including the altered balance in ccRCC toward excessive storage in the form of lipids. FA metabolism via ER-bound enzymes leads to production of diacylglycerol, which can then be stored as triglycerides in the lipid droplet; while FA transport into the mitochondrion via CPT1

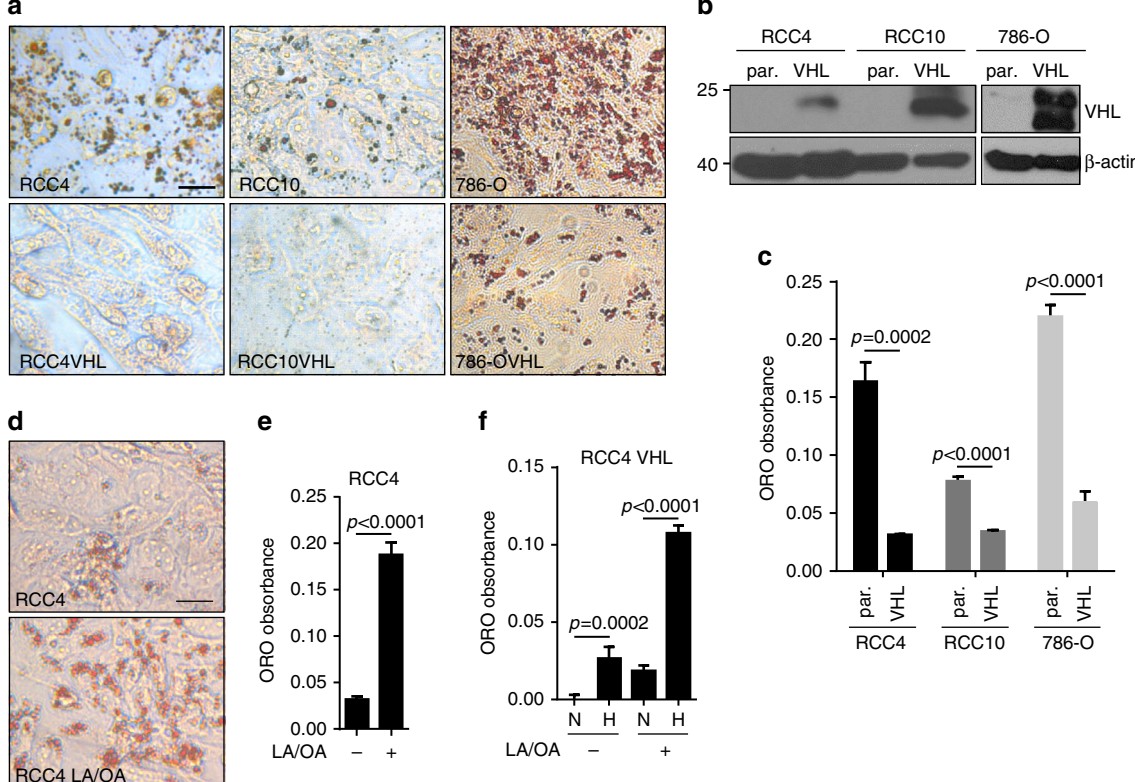

**Fig. 1** Lipid accumulation in ccRCC cells is VHL dependent. **a** Photomicrographs of RCC4, RCC10, and 786-O renal cell lines with or without VHL stained with Oil Red O 6 days after reaching confluence. **b** Western blot demonstrating VHL expression in reconstituted lines. β-actin used as loading control. **c** Quantification of Oil Red O extracted from cells shown in **a**. **d** Oil Red O staining of RCC4 cells 2 days after treatment with linoleic/oleic acid (LA/OA). **e** Quantification of Oil Red O staining of cells in **d**. **f** Quantification of Oil Red O staining of RCC4 VHL cells treated with LA/OA cultured in either normoxia (N) or 1% oxygen (H). Statistical tests for all panels were two-tailed Student's $t$ tests. Scale bar = 10 μm. Error bars represent standard deviations

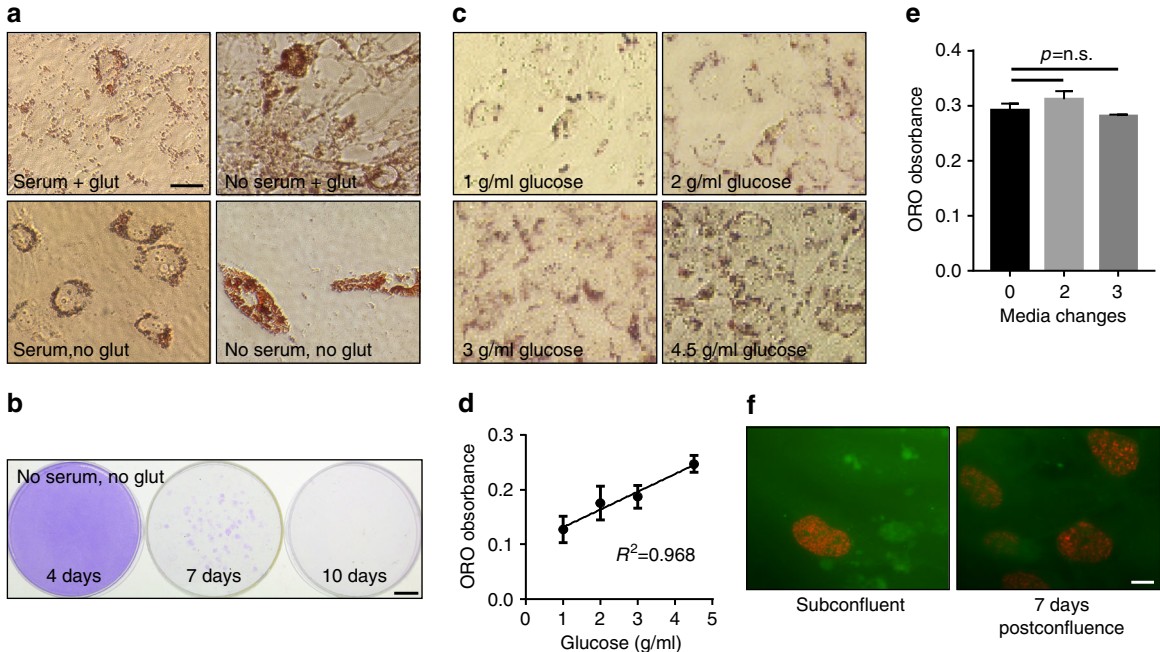

Fig. 2 Lipid accumulation is dependent on glucose, but independent of glutamine. **a** Photomicrographs of Oil Red O-stained 786-O cells cultured in the presence or absence of serum and glutamine, as indicated. Bar = 10 μm. **b** Crystal violet staining of rescued cells from treatment as in **a**. Bar = 1 cm. **c** Photomicrographs of Oil Red O-stained 786-O cells cultured in indicated glucose concentrations. **d** Quantification of Oil Red O of cells in **c** and graphed versus glucose concentration. Square of the Pearson correlation coefficient $r$ is shown. **e** Quantification of Oil Red O of 786-O cells with media changes 0, 2, or 3 times during the 7-day period. **f** Fluorescent labeling of BrdU incorporation of 786-O cells at subconfluent or at 7 days post confluence. Bar = 10 μm. $p$ values of two-tailed Student's $t$ tests are displayed. Error bars represent standard deviations

leads to beta oxidation and the regeneration of acetyl-CoA for entry into the critic acid cycle and the generation of reducing equivalents for ATP.

The goal of the current study was to determine the molecular mechanisms driving lipid deposition in ccRCC. To this point, it has remained unclear whether accumulation of lipids is a byproduct of altered metabolism in ccRCC, or whether lipid storage contributes to disease development. Here we define the rate-limiting enzyme of the FA transport system controlling entry into the mitochondrion, carnitine palmitoyltransferase 1A (CPT1A), as a hypoxia-repressed target gene that regulates lipid accumulation in ccRCC. CPT1A is shown to be a direct target gene of the HIF1 and HIF2 complexes and repressed in a VHL-dependent manner in ccRCC cells leading to reduced FA catabolism. Reintroduction of CPT1A into VHL-defective cells not only reverses the lipid deposition phenotype, but importantly reduces tumor growth in vivo. Analyses of clinical samples confirm the decreased expression and activity of CPT1A in renal tumors compared to normal kidney samples. Together, the findings highlight the importance of altered lipid metabolism in contributing to ccRCC, and suggest that targeting lipid metabolism in may be a new avenue for therapeutic intervention in renal cancer.

## Results

**Lipid deposition in ccRCC is VHL mediated**. To begin to dissect the mechanism of lipid deposition in ccRCC cells, we evaluated the ability of three ccRCC cell lines (RCC4, RCC10, and 786-O) to make lipid droplets. While others have found that ccRCC tumors demonstrate an adipogenic gene expression signature, and an ability to differentiate in vitro under an adipogenic differentiation protocol[8], we reasoned that terminal differentiation that is not characteristic of renal tumors might not be the only way to induce lipid deposits in ccRCC cells. Instead, we evaluated

the formation of lipid droplets in confluent monolayers. We found that all three cell lines developed lipid droplets as revealed by Oil Red O staining (Fig. 1a, upper panels) at 2–8 days after reaching confluence. In contrast, VHL reconstituted RCC4, RCC10, and 786-O cells (Fig. 1b) displayed a dramatic reduction in lipid droplet formation under the same conditions (Fig. 1a, lower panels). Quantification of the lipids after Oil Red O extraction normalized to cell number demonstrated that RCC4 cells accumulated greater than five times the lipid deposits compared to RCC4 VHL cells (Fig. 1c); RCC10 cells had greater than 2.25 times the lipids than RCC10 VHL cells; and 786-O cells accumulated 3.7-fold more lipids than 786-O VHL cells. Thus, ccRCC cells can indeed accumulate lipid droplets in vitro, as ccRCC tumors are characterized to do in vivo, without undergoing adipogenic differentiation. Moreover, lipid deposition in ccRCC cells depends on loss of the VHL tumor suppressor.

We next tested whether lipogenesis could be induced by treatment of ccRCC cells with cis-unsaturated fatty acids like linoleic (18:2) and oleic (18:1) acid (LA/OA) in subconfluent cells. LA/OA treatement has long been known to induce lipid droplets in adipocytes, but not kidney epithelial cells[12]. Indeed, we observed that RCC4 cells treated with 200 μM LA/OA, which is equivalent to the levels circulating in people consuming the western diet[13], more rapidly formed lipid droplets than untreated cells (Fig. 1d). Quantification of lipid droplet accumulation revealed greater than seven times the accumulation after 2 days of treatment in the LA/OA-treated RCC4 cells compared to the untreated cells (Fig. 1e). Finally, we tested whether the formation of droplets could be induced in RCC4 VHL cells when the cells were cultured under hypoxic conditions, and found hypoxia could potently induce lipid deposition in the presence or absence of LA/OA (Fig. 1f), suggesting a model that by regulating the hypoxia pathway, VHL negatively regulates lipid accumulation in renal cancer cells.

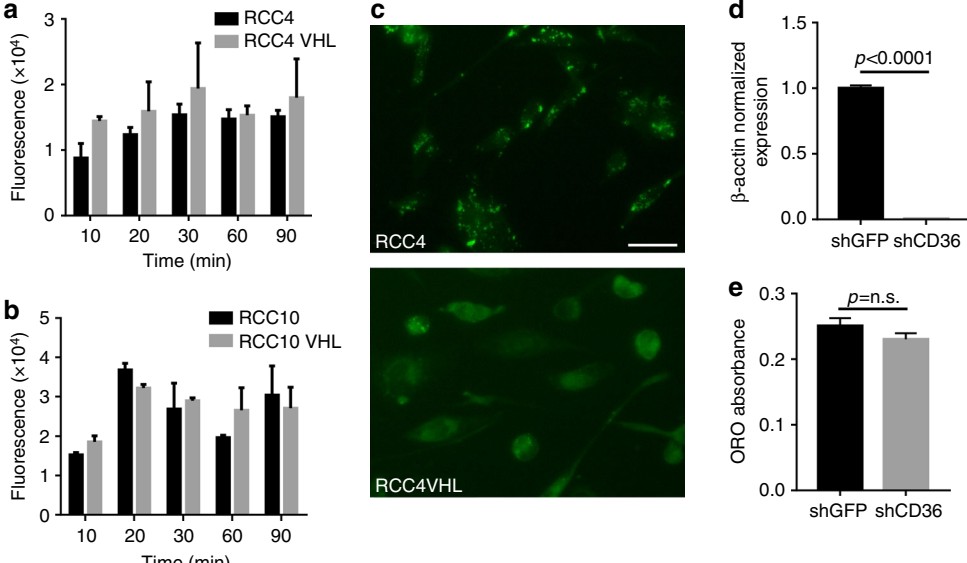

**Fig. 3** Lipid deposition in VHL-deficient cells is not due to altered fatty acid import. **a** Quantification of fluorescence in RCC4 and RCC4 VHL cells at indicated times after exposure to BODIPY-dodecanoic acid. **b** Quantification of fluorescence in RCC10 and RCC10 VHL cells at indicated times after exposure to BODIPY-dodecanoic acid. **c** Representative fluorescent images of BODIPY-labeled RCC4 and RCC4 VHL cells at 24 h after exposure to BODIPY-dodecanoic acid. Bar = 10 μm. **d** Quantitative real-time PCR detection of *CD36* expression after shRNA knockdown. **e** Quantification of Oil Red O staining of RCC4 cells with shGFP or shCD36 knockdown. Error bars represent standard deviations. *p* values of two-tailed Student's *t* tests are displayed

**Lipid droplet formation requires glucose, but not glutamine**. Recent studies have described vast rewiring of metabolic pathways in renal cell carcinomas to promote the Warburg effect, involving the metabolism of glutamine in reverse carboxylation to produce lipids and energy in place of the citric acid cycle in the mitochondria[10, 11]. We thus wondered if lipid deposition was dependent on glutamine metabolism, and tested droplet formation in 786-O cells in the presence or absence of glutamine. Cells were plated at confluence and observed for 7 days, and found to robustly form droplets in a glutamine-independent manner (Fig. 2a). Recent reports have demonstrated the dependence of ccRCC cells on glutamine as a primary carbon source for survival. Notably, very few cells were still attached to the surface of the plates, but the cells that survived demonstrated significant lipid droplet formation. To rule out the possibility that glutamine was present in the charcoal-stripped serum with which the medium was supplemented, we also performed the assay in the absence of both serum and glutamine and found similar results, and also noted that the remaining cells remained viable and maintained clonogenic capacity depending on how long they were starved (Fig. 2b). In contrast, droplet formation was highly dependent on the presence of glucose (Fig. 2c, d). Glucose concentration produced a dose-dependent increase in Oil Red O staining, occurring in the physiologic glucose range (1 g/L) as well as the standard culture range (4.5 g/L), and demonstrated an almost perfect correlation between glucose concentration and lipid droplet formation (correlation coefficient $R^2$ of 0.968). Droplet formation was found to be unrelated to nutrient exhaustion because changing the media during the course of deposition at either 2- or 3-day intervals did not diminish the production (Fig. 2e), and interestingly proliferation state revealed by BrdU labeling did not change up to 7 days post confluence, suggesting that droplet formation is not a postmitotic phenomenon (Fig. 2f). Together, the observations suggest lipid droplet formation in ccRCC cells is highly dependent on glucose rather than glutamine and is not a result of adverse culture conditions.

**Lipid uptake is not altered by VHL status**. We next assessed whether the differences between VHL competent and deficient cells in lipid deposition were due to differences in fatty acid uptake. A variety of fatty acid transporters, including CD36 and LRP1, have ties to hypoxia in other model systems, and could be involved in a VHL-dependent response in RCC[14–16]. To determine if lipid uptake is altered in VHL-deficient RCC cells, we used a fluorescent cis-unsaturated fatty acid molecule (BODIPY 500/510 C1, C12 (4,4-difluoro-5-methyl-4-bora-3a,4a-diaza-S-indacene-3-dodecanoic acid)), and monitored fatty acid uptake visually over time. As can be seen in Fig. 3a, b, VHL status had no effect on lipid uptake rates in RCC4 or RCC10 cells over a 90-min period as measured by fluorescent quantification. In contrast, fluorescent images at 24 h demonstrated significant differences in lipid localization between the VHL-expressing cells and the parental VHL-deficient cells (Fig. 3c). In agreement with the Oil Red O staining, BODIPY was localized in foci reminiscent of lipid droplets in the RCC4 cells, but was diffuse in the RCC4 VHL cells. To specifically rule out CD36, an hypoxia-inducible scavenger receptor for oxidized low-density lipoprotein and long-chain fatty acids[14], as a mediator of lipid uptake, we performed shRNA knockdown of CD36 in RCC4 cells (Fig. 3d) and found that almost complete knockdown had non-significant effects on lipid droplet formation (Fig. 3e). Thus, lipid uptake appears to be unable to explain the differences in droplet formation; rather, lipid catabolism appears altered.

**Lipid droplet formation is dependent on HIF**. As HIF is the major mediator of VHL-dependent functions, we investigated the roles of HIF1α and HIF2α in lipid accumulation in ccRCC cells. Both HIF isoforms were inactivated by shRNA in RCC4 cells either alone or in combination, as demonstrated by western blot in Fig. 4a, and the cells were tested for lipid accumulation post confluency. Knockdown of either HIF isoform resulted in a statistically significant decrease in droplet formation compared to

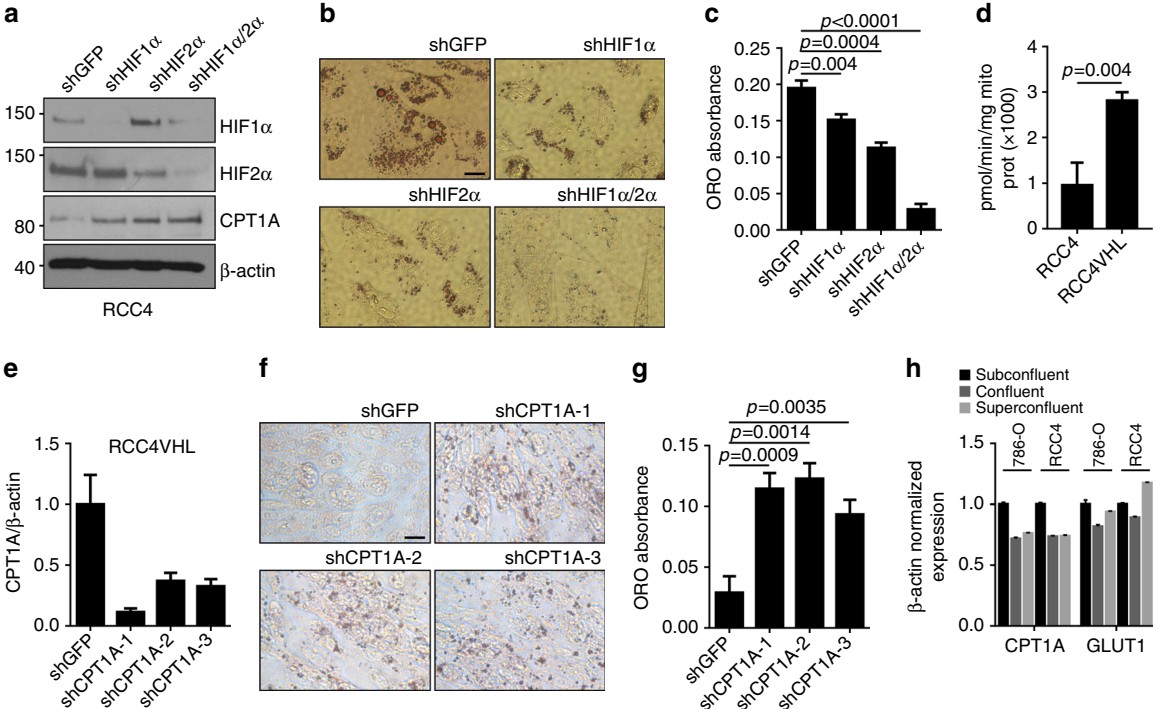

**Fig. 4** CPT1A repression controls lipid deposition in ccRCC cells. **a** Western blot of RCC4 cell lysates expressing shRNA to HIF1α and HIF2α decorated with antibodies to HIF1α, HIF2α, CPT1A, and β-actin. **b** Photomicrographs of RCC4 cells with HIF1α and HIF2α knockdown stained for Oil Red O. **c** Quantification of Oil Red O staining of cells in **b**. **d** CPT1A activity measurement in RCC4 and RCC4 VHL mitochondria. **e** Quantitative real-time PCR of *CPT1A* (qRTPCR) expression in RCC4 VHL cells expressing three different CPT1A shRNA constructs. **f** Photomicrographs of Oil Red O-stained RCC4 VHL cells with *CPT1A* knockdown. Bar = 10 μm. **g** Quantification of cells in **e**. **h** Quantification of *CPT1A* and *GLTU1* expression in subconfluent, confluent, or superconfluent 786-O and RCC4 cells. Error bars represent standard deviations. *p* values of two-tailed Student's *t* tests are displayed

the shGFP control cells, ranging from 23% for the HIF1α knockdown to 42% for the HIF2α knockdown (Fig. 4b, c; *p* = 0.004 and 0.0004, respectively, Student's *t* tests). The most dramatic decrease resulted from the double knockdown of HIF1α and HIF2α (*p* < 0.0001, Student's *t* test), demonstrating the HIF dependence of lipid storage in RCC cells.

Lipid synthesis begins ostensibly as excess citrate accumulates in the cytosol is converted to acetyl-CoA and becomes a substrate for acetyl-CoA carboxylase and subsequently fatty acid synthase[9]. Fatty acids are then activated by acyl-CoA synthase, and are either shuttled back into the mitochondria after modification by carnitine palmitoyltransferase 1 (CPT1A) on the mitochondrial outer membrane, or are further modified into di- and triglycerides for storage in the lipid droplet. CPT1A is the rate-limiting step in determining FA oxidation or storage. Because we observed lipid droplet formation in ccRCC cells in a glucose or FA-dependent manner, we investigated steps downstream from FA anabolism. We first measured the levels of CPT1A in RCC4 cells depleted for HIF1α, HIF2α, or both, and found expression of CPT1A to be inversely related to the levels of either HIF1α or HIF2α (Fig. 4a), in agreement with a model of increased HIF in response to VHL inactivation leading to decreased CPT1A and subsequently increased lipid storage. We measured the activity of CPT1A in RCC4 and RCC4 VHL mitochondria and found a correlated increase in relative activity in the presence of VHL (Fig. 4d). To determine directly the effect of CPT1A depletion on droplet formation, we used shRNA to reduce *CPT1A* message (Fig. 4e), and then monitored lipid droplet formation in RCC4 VHL cells. As shown in Fig. 4f, all three knockdown constructs dramatically increased lipid deposition compared to the control RCC4 VHL shGFP cells, in the range of three-fourfold (Fig. 4g). Lastly, we asked whether confluence influences CPT1A expression since we

noted that lipid accumulation occurs more robustly at post-confluent conditions and observed a modest but reproducible decrease in both the RCC4 and 786-O cells that appeared to be HIF independent because no correlating changes were seen in *GLUT1* (Fig. 4h). Thus, CPT1A repression by the HIF pathway mediates the effects of VHL loss regarding lipid storage.

**HIF1 and HIF2 bind the *CPT1A* locus and inhibit expression**. To delineate how HIF regulates CPT1A, we assessed the effect of hypoxia on *CPT1A* message and protein levels in RCC4 and RCC4 VHL cells. Introduction of VHL into RCC4 cells led to a >fourfold increase in *CPT1A* mRNA (Fig. 5a), while treatment of RCC4 VHL cells with 1% oxygen for 24 h reversed the effect almost completely. At the protein level, similar results were observed. Introduction of VHL led to a drop in HIF1α levels, and an inversely coordinated increase in CPT1A; both of which were reversed by hypoxia exposure (Fig. 5b). Because HIF is not typically considered a direct transcriptional repressor[17], we next assessed the contribution of two transcriptional repressors that are induced by HIF stabilization, Snail and DEC1, that from the literature might be predicted to be involved in metabolic changes. DEC1 has recently been identified as the mechanism through which HIF suppresses PGC1α and reduces mitochondrial respiration[18]; while Snail affects glucose metabolism[19]. Unexpectedly, however, knockdown of neither Snail (Fig. 5c) nor DEC1 (Fig. 5d) in RCC4 cells was able to alleviate the suppression of CPT1A, in spite of elevated expression of E-cadherin (Snail target[20]) or PGC1α (DEC1 target[18]), respectively. Conversely, knockdown of CPT1A did lead to a reduction in PGC1α expression suggesting some overlap of the pathways (Fig. 5e), but the data clearly distinguish CPT1A from DEC1. *PLIN2*, another

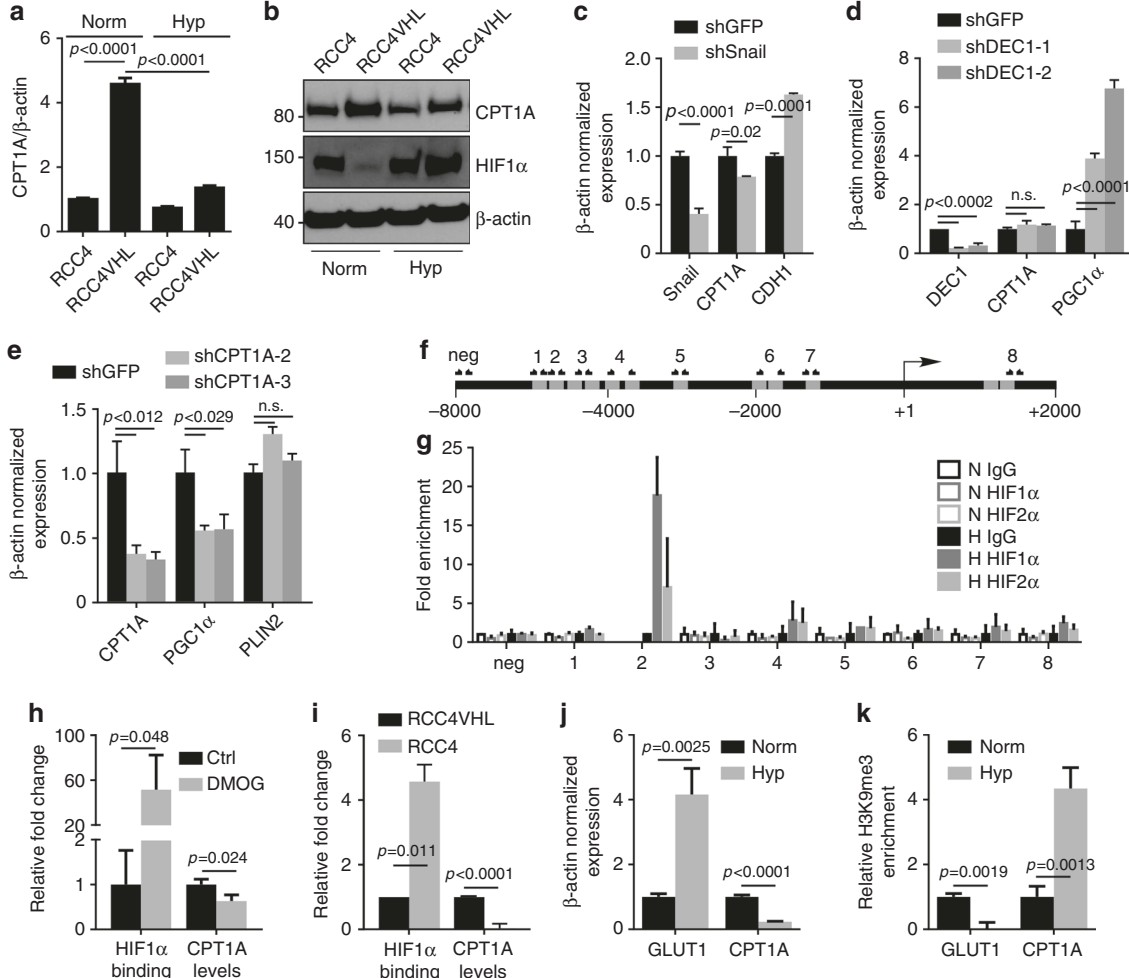

**Fig. 5** HIF1 and HIF2 bind *CPT1A* and inhibit expression. **a** mRNA expression of *CPT1A* in RCC4 and RCC4 VHL cells in normoxia or hypoxia as measured by qRTPCR normalized to β-actin. **b** Western blot depicting protein expression of CPT1A, HIF1α, and β-actin in RCC4 cells in normoxia or hypoxia for 24 h. **c** Quantification of the effect of shRNA knockdown of Snail in RCC4 cells on *CPT1A* and *CDH1* (*E-cadherin*) expression. **d** Quantification of the effect of shRNA knockdown of DEC1 with two different shRNAs in RCC4 cells on *CPT1A* and *PGC1α* expression. **e** Quantification of the effect of shRNA knockdown of CPT1A in RCC4 cells on *PGC1α* and *PLIN2* expression. **f** Diagram of the *CPT1A* promoter region analyzed for putative HREs (gray boxes) 8000 base pairs upstream to 2000 base pairs downstream of the transcriptional start site (+1). Primer pairs used for PCR amplification after ChIP are indicated. **g** qRTPCR results using primers pairs indicated in **f** of ChIP with antisera to HIF1α or HIF2α performed on RCC4 VHL chromatin after treating the cells with normoxia (open boxes) or hypoxia (closed boxes) for 24 h. **h** HIF1α ChIP on the CPT1A region 2 HRE in lysates of RCC4 VHL cells treated with DMOG for 36 h ("HIF1α binding") compared with CPT1A RT-PCR ("CPT1A levels"). **i** HIF1α ChIP on the CPT1A region 2 HRE in lysates of normoxic RCC4 and RCC4 VHL cells. **j** mRNA expression of GLUT1 and *CPT1A* in RCC4 VHL cells in normoxia or hypoxia as measured by qRTPCR normalized to β-actin. **k** qRTPCR results of histone H3 lysine 9 trimetylation ChIP on the GLUT1 and *CPT1A* promoters of RCC4 VHL cells in normoxia or hypoxia. Error bars represent standard deviations. *p* values of two-tailed Student's *t* tests are displayed

recently described HIF target involved in lipid droplet biology[21] also appears unrelated to *CPT1A* regulation (Fig. 5e). Thus, the mechanism of regulation of *CPT1A* by HIF appears distinct from other known metabolic effectors.

We next examined the 10 kilobases surrounding the transcriptional start site of *CPT1A* to see if potential HIF response elements (HRE) could be found bioinformatically. MatInspector analysis (Genomatix.com) revealed 12 putative HREs throughout the region (Fig. 5f). We designed eight primer pairs specific for each HRE region, grouping those that fell within 50–100 base pairs of each other, and performed chromatin immunoprecipitation (ChIP) with either HIF1α or HIF2α-specific antibodies. RCC4 VHL cells were treated with 1% oxygen for 24 h, and then harvested for ChIP. We found that of the eight primer pairs, region 2 was enriched in both the HIF1α and HIF2α pull-down experiments after hypoxia treatment, suggesting HIF1α or HIF2α

can both bind specifically to the *CPT1A* promoter in a hypoxia-dependent manner (Fig. 5g). We tested whether HIF1α could bind and regulate CPT1A under normoxic conditions in RCC4 VHL cells by treating the cells with the HIF prolyl-hydroxylase inhibitor dimethyloxalylglycine, N-(methoxyoxoacetyl)-glycine methyl ester (DMOG, 1 mM, 36 h); and in RCC4 cells which have high basal HIF1α due to the lack of VHL. We found both an increase in HIF1α association and a concurrent decrease in CPT1A message under normoxia in both situations, suggesting HIF activation is sufficient for the effect rather than requiring additional hypoxia-mediated changes (Fig. 5h, i). We then asked whether hypoxic regulation of the *CPT1A* promoter altered epigenetic marks near the start site that are associated with expression versus silencing. ChIP of RCC4 VHL cells treated with hypoxia demonstrated that an increase in GLUT1 mRNA (Fig. 5j) was associated with a decrease in histone H3 lysine 9

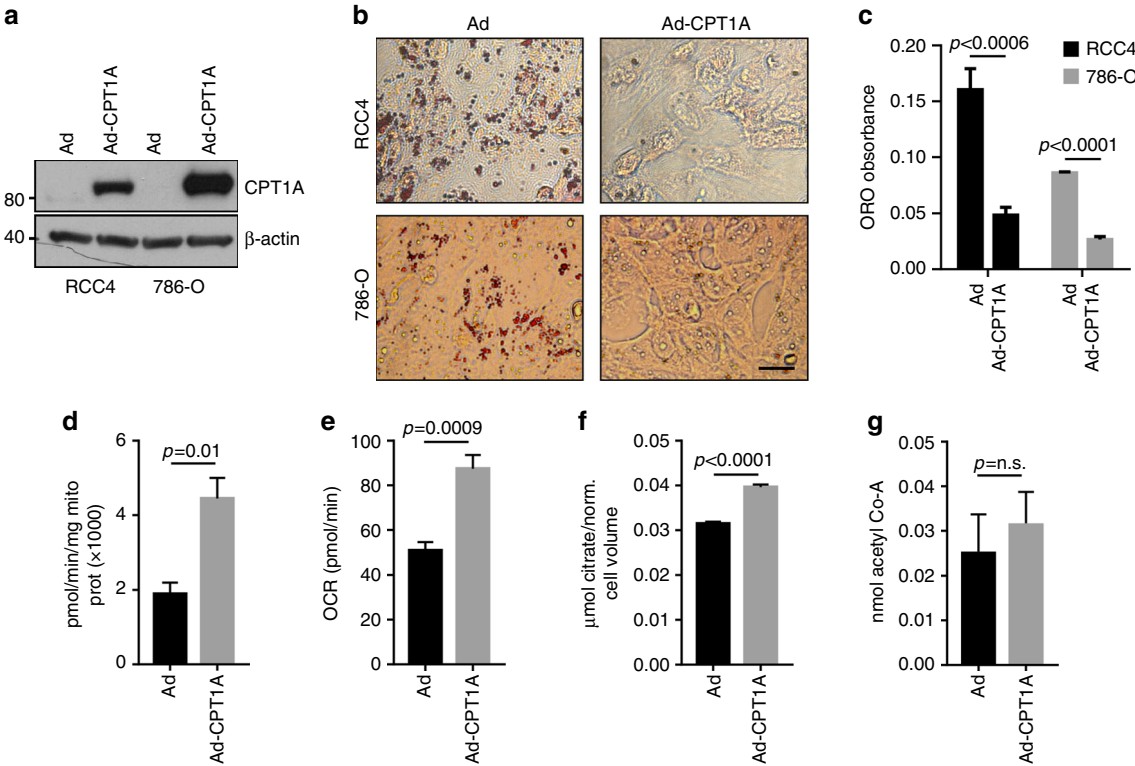

**Fig. 6** CPT1A expression prevents lipid deposition and alters mitochondrial function in ccRCC cells. **a** Western blots of RCC4 and 786-O cell lysates after adenoviral infection with CPT1A of GFP-expressing adenovirus stained for CPT1A or β-actin. **b** Photomicrographs of Oil Red O staining of RCC4 or 786-O cells after infection with GFP or CPT1A adenovirus. Bar = 10 μm. **c** Quantification of Oil Red O staining depicted in **b**. **d** CPT1A activity measurement in adenovirally infected RCC4 cells. **e** Oxygen consumption rate (OCR) of adenovirally infected RCC4 cells. **f** Citrate levels in adenovirally infected RCC4 cells. **g** Acetyl-CoA levels adenovirally infected RCC4 cells. Error bars represent standard deviations. *p* values of two-tailed Student's *t* tests are displayed

trimethylation (H3K9me3) of the GLUT1 promoter (Fig. 5k). Accordingly, hypoxia led to a decrease in *CPT1A* mRNA and a robust increase in trimethylation of lysine 9. Together, we conclude that hypoxia leads to a silencing of *CPT1A* transcriptionally in a HIF-dependent, non-canonical manner.

**CPT1A restoration blocks lipid accumulation and tumorigenesis.** While lipid deposition has long been a defining characteristic of ccRCC, the significance of the change in lipid metabolism to tumorigenesis remains unclear. To assess the effects of restoring CPT1A levels in ccRCC cells, we infected RCC4 and 786-O cells with an adenovirus encoding a constitutively active form of rat CPT1A that is not inhibited by malonyl-CoA (called CPT1AM) and assessed changes to lipid deposition and tumor growth. Adenoviral transduction led to dramatic increases in expression of CPT1A (Fig. 6a), and associated decreases in the ability of both RCC4 and 786-O cells to produce droplets (Fig. 6b). Quantification of droplet formation confirmed this observation (Fig. 6c). We confirmed the function of CPT1A by testing activity in the infected cells and found a greater than twofold increase (Fig. 6d), similar to the change seen in RCC4 compared to RCC4 VHL cells (Fig. 4d). In accordance with increased lipid oxidation, elevation of CPT1A increased oxygen consumption (Fig. 6e) and citrate levels (Fig. 6f), in line with changes observed in ccRCC due to VHL loss[11]. Acetyl-CoA levels, however, did not change appreciably (Fig. 6g), perhaps due to compensatory mechanisms.

We next assessed the ability of CPT1A-restored cells to form tumors in nude animals; 786-O cells were injected subcutaneously into the flanks of nude mice, and followed over a period of 6 weeks. CPT1A restoration led to significant decreases in tumor growth, as

assessed by two-way ANOVA ($p = 0.0034$) (Fig. 7a). Notably, we extracted protein lysates from a number of tumors, and observed that all of the CPT1A-expressing tumors had lost expression by the end of the assay (Fig. 7b), which is not surprising due to the transient nature of adenoviral gene transfer. In contrast, one tumor that was harvested at the size of 60 mm³ was stained for CPT1A expression by immunofluorescence and with Oil Red O for lipid droplet detection revealed abundant CPT1A and sparse lipid deposition compared to a control tumor (Fig. 7c).

We next wanted to determine whether stable expression of CPT1A could lead to more profound effects on tumor growth, and thus turned to creating a doxycycline-inducible cell line. 786-O cells were therefore transfected with a doxycycline regulatable vector, and clonal lines were established (Fig. 7d). Of numerous clones, the clone that exhibited the highest level of CPT1A induction after 24 h of doxycycline administration was chosen (clone #14). To verify that lipid accumulation was affected by CPT1A expression in this subclone, cells were first tested for droplet formation in vitro in the presence of doxycycline or vehicle (dimethylsulfoxide (DMSO)). As shown in Fig. 7e, doxycycline was able to dramatically reduce lipid accumulation in the cells, similar to the effects of adenoviral CPT1A expression in the parental cells in Fig. 6c. Subsequently, clone #14 cells were implanted subcutaneously in nude animals and tumor growth was monitored. When average tumor volumes reached 100 mm³, the animals were randomized and doxycycline was added to the chow of one cohort of mice. Over the next 5 weeks, it became evident that animals eating normal chow developed tumors at a significantly increased rate compared to the animals ingesting doxycycline ($p < 0.0001$, two-way ANOVA) (Fig. 7f). Thus, we conclude that CPT1A expression is incompatible with ccRCC

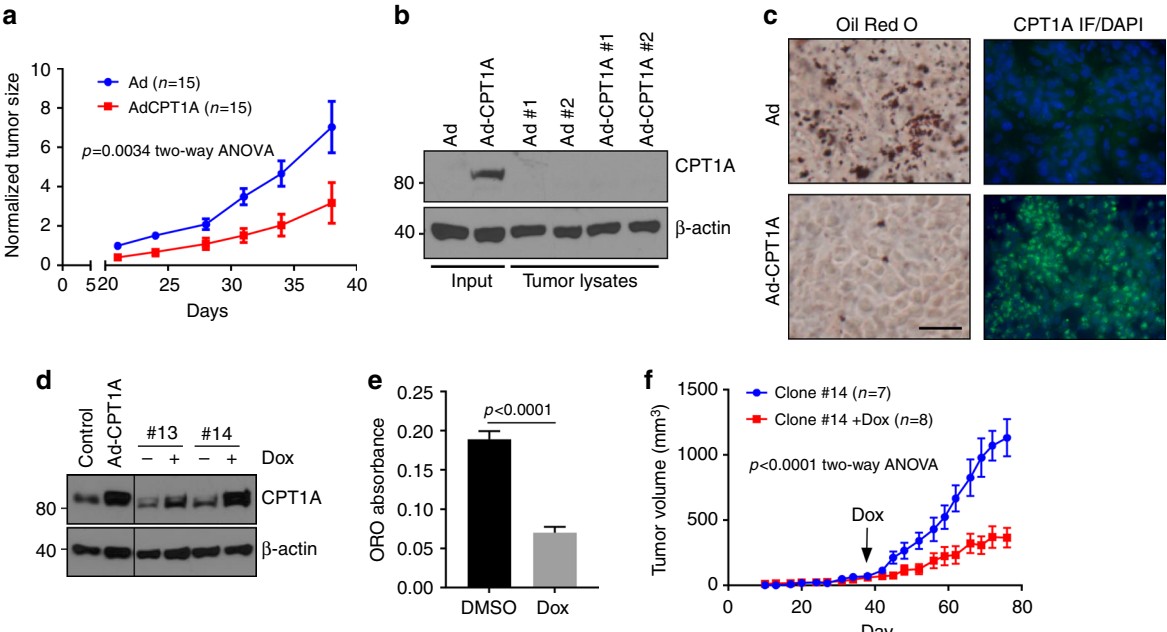

**Fig. 7** CPT1A expression limits tumor growth of ccRCC cells in vivo. **a** Tumor growth measurements of 786-O cells infected with GFP or CPT1A-expressing adenovirus implanted on the flanks of nude mice. **b** Western blot of various protein lysates extracted from tumors measured in **a** stained for CPT1A and β-actin levels. **c** Photomicrographs of Oil Red O staining and immunofluorescent staining of a GFP and CPT1A tumor described in **a**. Immunofluorescence was performed with a FITC-labeled secondary antibody and counterstained with DAPI. Bar = 20 μm. **d** Western blot of depicting doxycycline-inducible CPT1A expression in two 786-O clonal cell lines compared to adenoviral-infected control cell lines. Blots were probed for CPT1A and β-actin. **e** Quantification of Oil Red O staining of clone #14 treated with or without doxycycline. **f** Tumor growth measurements of clone #14 implanted subcutaneously into the flanks of nude mice. Doxycycline food was given to the Dox cohort when average tumor volumes reached ~100 mm³ through to the end of the assay. Error bars represent standard deviations. *p* values for two-way ANOVAs for **a**, **f**, and a two-tailed Student's *t* test (**e**) are displayed

tumor growth, and that HIF-mediated suppression of CPT1A is a requisite step in ccRCC development in model systems.

**CPT1A expression and activity are repressed in ccRCC samples.** To understand the clinical significance of CPT1A suppression in ccRCC, we queried cancer gene expression databases for *CPT1A* levels in patient samples. We found statistically significant reductions of *CPT1A* message in tumors compared to normal kidney specimens in both the Beroukhim[22] and Gumz[23] studies ($p < 0.0001$ and $p = 0.0003$, respectively; Student's *t* tests; Fig. 8a). We next wished to determine if *CPT1A* expression associates with patient outcome, and thus interrogated The Cancer Genome Atlas (TCGA) database[2]. As shown in Fig. 8b, division of patients into upper and low thirds by expression level resulted in a significant difference in the likelihood of survival in favor of those with higher expression ($p = 2.58 \times 10^{-6}$, Log-rank test), implying the level of suppression of CPT1A is related to the severity of disease. To determine if differences in gene expression correlate with differences in enzymatic activity, we measured CPT1 activity in the mitochondria of freshly isolated ccRCC tumors and compared to activity in mitochondria from matched normal kidney specimens. In seven samples, we found an average decrease in activity of 81% in validated VHL mutant tumors compared to their normal tissues, and found that CPT1A was repressed in all cases (Fig. 8c, d). We also found robust lipid droplet formation in the tumors (Fig. 8e). Together, the studies indicate that CPT1A repression is recapitulated in human tumor samples, and that the level of repression may have prognostic value.

## Discussion

In the current work, we present data demonstrating that metabolic adaptation in clear cell renal cell carcinoma resulting in

activation of lipid storage pathways is a necessary step in the development of malignancy. We identify the rate-limiting enzyme in fatty acid entry into the mitochondria (CPT1A) as a direct target of transcriptional repression by the HIF1α and HIF2α subunits through promoter silencing. Re-expression of VHL in ccRCC cell lines results in degradation of HIF1α and HIF2α, elevation of CPT1A message and protein, and inhibition of the formation of lipid droplets. Accordingly, depletion of CPT1A in VHL reconstituted cell lines restores the lipid deposition phenotype. The importance of CPT1A repression was determined in re-expression studies, in which we observed an inhibition of lipid deposition and a correlated inhibition of tumor growth. Finally, we observed that in clinical samples, CPT1A expression is reduced in tumors compared to normal kidneys, and significantly that CPT1A activity is reduced in mitochondria harvested from tumor tissue compared to adjacent normal kidney tissue. Thus, CPT1A expression is a critical target of the HIF pathway involved in clear cell tumor etiology.

Altered metabolism is a well-recognized hallmark of cancer[24], and ccRCC offers perhaps one of the most visually convincing examples. Metabolic changes in ccRCC include shifts to anaerobic metabolism through the HIF-dependent activation of many genes in the glycolytic pathway, reductive carboxylation of glutamine-derived α-ketoglutarate as a carbon source in place of pyruvate in the citric acid cycle, and increased utilization of the pentose phosphate pathway[11, 18, 25–27]. The notable change histologically, however, is the appearance of vast lipid and glycogen deposits that fill the cytoplasm of tumor cells[28]. It has been unclear until recently whether lipid deposition in cancer cells promotes a selective advantage to cancer cells, or whether lipid storage is a passenger to altered energy metabolism. The Harris group, however, demonstrated that hypoxia-induced-lipid storage in breast and glioma cell lines is due to a HIF-dependent increase in

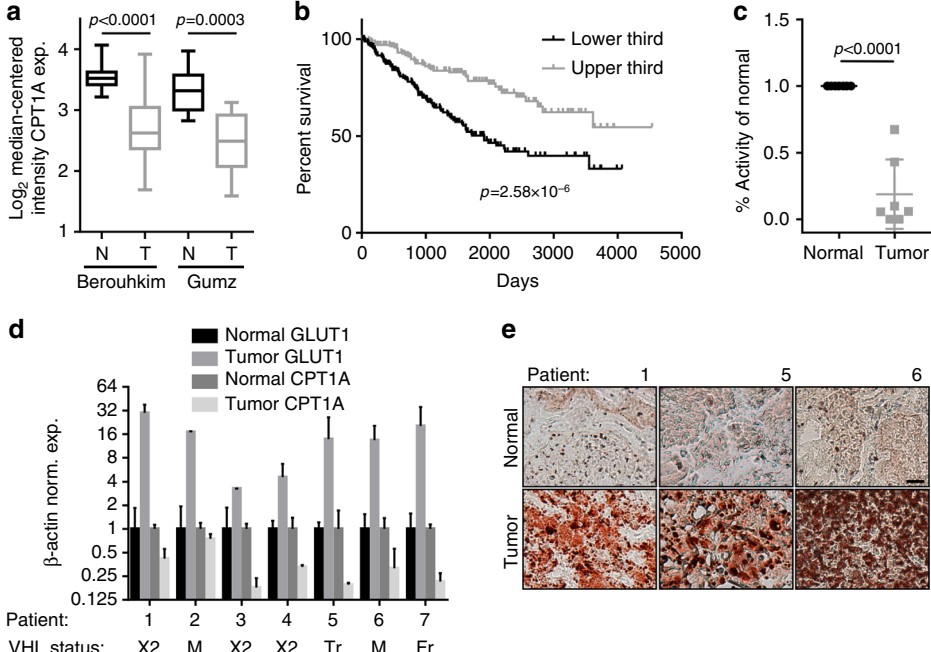

**Fig. 8** CPT1A expression and activity are reduced in ccRCC tumor tissue. **a** Box and whisker plots of Oncomine data demonstrating decreased expression of *CPT1A* mRNA in two studies. Data are depicted as Log₂ transformed, median-centered intensity expression of *CPT1A* mRNA in normal (N) versus tumor (T). In the Berouhkim data set, *CPT1A* is repressed 1.82-fold; in the Gumz data set, *CPT1A* is repressed 1.79-fold. **b** Kaplan–Meier survival plot of patients with upper and lower third expression of *CPT1A* from the TCGA ($n = 172$ for each group). **c** Activity measurements of CPT1 in mitochondria isolated from clear cell renal carcinoma tumors versus adjacent normal tissues in seven patients. Data are presented as % change compared to the activity in matched normal tissues. **d** Normalized expression of *GLUT1* and *CPT1A* in tumor and normal tissues of ccRCC patient samples from **c**, as well as VHL status indicated below (×2 = exon 2 deletion, M = mutation, Tr = truncation, Fr = frame shift). **e** Representative Oil Red O staining of three patient tumor samples and normal kidneys, as indicated. Bar = 20 μm. *p* values of two-tailed Student's *t* tests are displayed (**a**, **c**) and Log-rank test (**b**)

fatty acid uptake via transcriptional regulation of fatty acid-binding proteins FABP3 and FABP7[29]. Importantly, interruption of the pathway decreased lipid droplets and the tumorigenic capacity of xenografts in mice while increasing cellular reactive oxygen species (ROS), leading the authors to conclude that hypoxia-driven lipid accumulation serves as a protective barrier against oxidative stress-induced toxicity. Similarly, it was recently published by the Simon group in clear cell renal tumors that lipid deposits offer protection from hypoxia or pharmacologically induced ER stress[21]. Both publications identify the lipid droplet structural protein perilipin2 (known as PLIN2 or ADRP) as hypoxia-regulated and critical for droplet formation.

Unlike other studies, however, the current work demonstrates a key metabolic enzyme is essential for the lipid deposition phenotype. CPT1A, part of the CPT1 transport system, is the rate-limiting component controlling the entry of fatty acids into the mitochondria for beta oxidation. CPT1 catalyzes the transfer of the acyl group from palmitoyl-Coenzyme A to carnitine to form palmitoylcarnitine, which is subsequently transported across the inner mitochondrial membrane in exchange for free carnitine by carnitine-acylcarnitine translocase. Under hypoxic conditions, decreasing the production of acetyl-CoA by beta oxidation in addition to decreasing production of acetyl-CoA via decarboxylation of pyruvate are logical adaptations to decreased oxygen availability. HIF is responsible for inducing the expression of *PDK1* (pyruvate dehydrogenase kinase 1) that inhibits pyruvate dehydrogensase[25], and likewise we find here that HIF also is responsible for inhibiting the expression of *CPT1A*. Together, HIF alters aerobic metabolism by diverting pyruvate to glycolysis, and fatty acids to storage.

HIF functioning as a transcriptional repressor is unusual but not undocumented[17]. The common way that hypoxia leads to

transcriptional repression is through the induction of transcriptional repressors including DEC1, DEC2, ZEB1, ZEB2, Snail, and REST. We tested two of the known repressors we thought might be most relevant to metabolic changes (DEC1 and Snail) based on prior publications suggesting DEC1 controls respiration[18] and Snail controls glucose metabolism[19], but were unable to implicate them in *CPT1A* regulation. Clearly further studies will be necessary to determine how HIF functions differently on *CPT1A* than on activated targets, such as by identifying proteins that co-localize with HIF on one promoter but not another. While our studies are by no means exhaustive in this regard, the binding of HIF to a repressed gene target that is negatively regulated strongly suggests the localization is mechanistically relevant. Notably, HIF activity in the case of *CPT1A* suppression is sufficient in the absence of other hypoxia-regulated epigenetic regulators, such as the oxoglutarate-dependent dioxygenases of the Jumonji family whose expression are not only induced by hypoxia, but require low oxygen for activity. Reduction of *CPT1A* message by HIF is seen in normoxia in VHL-deficient cells as well as in DMOG-treated cells in normoxia. A significant question that remains open is whether HIF localizes with a co-repressor on *CPT1A*, and how the distinction is made between activation or inhibition of a HIF transcriptional target.

Cancer cells frequently display alterations in lipid metabolism as carbon demand for the formation of membranes and signaling molecules is high in rapidly dividing tumor cells[9]. It has been noted that while normal cells rely on fatty acid uptake as a major source of fatty acids, tumor cells prefer to synthesize their fatty acids[30]. We tested whether the elevated storage of fatty acids in ccRCC tumor cells was due to increased uptake from extracellular sources using a fluorescently labeled cis-unsaturated fatty acid, but found accordingly that uptake of the labeled fatty acid was not altered. We also tested the role of the lipid transporter CD36,

which is known to be hypoxia regulated[14]. Unlike the published observations for hypoxic MCF7 and U87 cells[29], therefore, the renal lines we tested do not develop lipid droplets due to increased lipid or fatty acid transport. We further found that glucose but not glutamine was required for the formation of droplets, which was surprising based on the recent findings regarding the use of glutamine by renal cancer cells for the production of citrate and subsequently lipogenesis[11]. We also noted the reported sensitivity of VHL-deficient renal cancer cells to glutamine starvation, but remarkably the cells were still able to store lipids in droplets. At present, it is unclear whether the requirement for glucose is for the production of glycerol to make triglycerides, or to make fatty acids via de novo synthesis. It is also possible that the surviving tumor cells in glutamine-deprived conditions acquire lipids from neighboring cells that have undergone cell death[31]. What is clear, however, is that the fate of fatty acids is drastically altered due to the repression of CPT1A expression and activity.

A recent integrative analysis of metabolomics and metabolic gene expression in ccRCC revealed a suppression of gluconeogenesis by uniform depletion of fructose-1,6-bisphosphatase-1 (FBP1) in a non-HIF-dependent manner[26]. By reducing FBP1, ccRCC cells promote glycolysis and in theory leave glycerol for use in lipid formation. Interestingly, a report describing a genetic screen for shRNAs that could increase survival of renal proximal epithelial cells (the proposed cell of origin for ccRCC) after treatment with severe hypoxia (0.2% $O_2$ for 48 h) identified both FBP1 and CPT1A as candidate genes whose suppression led to survival[32]. Thus, as both FBP1 and CPT1A are inhibitive of tumor growth and cell survival under stress, both genes display tumor suppressor phenotypes; though whether lipid deposition is a common mechanism of the effect remains to be investigated. Importantly, the discoveries surrounding the roles of these and other genes in ccRCC suggest that pharmacologic approaches to reverse the lipid deposition phenotype could have traction in treating renal cancer. A similar argument has been made recently in a mouse model of renal fibrosis leading to kidney disease, wherein it was observed that fatty acid oxidation is diminished and lipid deposition is elevated[33]. Restoring fatty acid metabolism with C75, a compound that induces CPT1 activity, led to reduction or prevention of the fibrotic phenotype. (Unfortunately, however, C75 has also been argued to inhibit CPT1 in other systems[34].) Together, the observations make the case for new therapeutics in renal cancer that target various aspects of the characteristic changes in metabolism, including restoration of function of FBP1 and CPT1A.

CPT1 may have a unique role in limiting tumor growth in ccRCC. Studies have shown that expression of CPT1C, the brain isoform of CPT1A, is elevated in lung cancer and promotes tumor growth and resistance to metabolic stress[35]. In addition, etomoxir, an irreversible CPT1 inhibitor increases reactive oxygen species and cell death in glioblastoma cells. Finally, leukemia cells are also sensitive to inhibition of fatty acid oxidation. In non-tumor cells, CPT1 has an important role in angiogenesis because fatty acid oxidation has been shown to be required for nucleotide synthesis for DNA replication[36]. Thus, our observation that CPT1A repression is critical for ccRCC growth identifies a tumor-specific metabolic adaptation of renal cancer and supports the concept that limiting FA availability by increasing oxidation could have clinical potential[9]. Efforts to develop specific inducers of CPT1A expression or activity may therefore be warranted as a novel therapeutic approach.

## Methods

**Cell culture and reagents**. All cell lines were STR (short tandem repeat) verified and mycoplasma tested. RCC4, RCC4 VHL (Sigma); RCC10, RCC10 VHL (gift of

Dr Amato Giaccia); and 786-O, 786-O VHL (gift of Dr Sandra Turcotte) cells were maintained in high-glucose DMEM medium supplemented with 10% bovine calf serum at 37 °C and 5% $CO_2$. For hypoxia incubation, a 1% $O_2$ environment was generated in a Ruskinn InVivo2 Hypoxia chamber. Lipofectamine 2000 (Life Technologies) was used for transfections. Crystal violet staining was performed with 0.05% crystal violet in 1% formaldehyde and 1% methanol. BrdU labeling was performed by supplementing cells with 0.03 mg/ml for 24 h, and staining with anti-BrdU (Cell Signaling) followed by Texas red-conjugated goat anti-mouse IgG. Non-specific FITC fluorescence was used as a background cell stain.

**Oil Red O staining**. Cells plated in 12-well plates at triplicate were rinsed with PBS twice, and fixed with 10% formaldehyde for 1 h. They were then rinsed with 60% isopropanol for 5 min, stained with 3 mg/ml Oil Red O for 4 min, and washed with water three times. For Oil Red O quantification, the cells were dried, and 250 µl of isopropanol was added and incubated for 3 min; the eluted solution was read at 510 nm. To normalize for cell mass, the cells were subsequently stained with sulforhodamine B following a published protocol[37], and read at 564 nm wavelength. Fresh frozen, OCT-embedded tumor sections were stained with a similar protocol after sectioning 8 µm slices on a cryostat.

**Real-time PCR**. TRIzol reagent was used for RNA isolation. RT-PCR was performed using standard procedures and normalized to β-actin. Briefly, RNA samples were reverse transcribed into cDNA using MMLV-RT, and quantitative PCR was performed with SYBR green master mix or TaqMan Fast Advanced Master Mix (Life Technologies) on the StepOne plus PCR system (Applied Biosystems). Primer sequences were β-actin F: 5′-CATGTACGTTGCTATCCAGGC-3′ R: 5′-CTCCTT AATGTCACGCACGAT-3′; CD36 TaqMan Hs01567187; CPT1A F: 5′-GAAGAT GGCAGAAGCTCACC-3′ R: 5′-TGGCGTACATCGTTGTCAT-3′; GLUT1 F: 5′-GGCCAAGAGTGTGCTAAAGAA-3′ R: 5′-ACAGCGTTGATGCCAGACAG-3′; Snail TaqMan Hs00195591; CDH1 TaqMan Hs1023895; DEC1 F: 5′-CCTTGA AGCATGTGAAAGCA-3′ R: 5′-GCTTGGCCAGATACTGAAGC-3′; PGC1α F: 5′-GTCAACAGCAAAAGCCACAA-3′ R: 5′-TCTGGGGTCAGAGGAAGAGA-3′; PLIN2 F: 5′-CCTGCTCTTCGCCTTTCG-3′ R: 5′-TGCAACGGATGCCATTT TT-3′.

**Western blots**. Western blots were performed using standard procedures. Briefly, protein lysates were created with a 9 M urea lysis buffer (7.5 mM Tris/HCl, pH 8.0), followed by sonication using a Sonic dismembrator and centrifugation at 10,000 rpm for 20 min. Protein concentrations were determined by the BCA (Bicinchoninic acid) method on a NanoDrop 1000. Between 50 and 100 µg of protein was electrophoresed on polyacrylamide gels, and transferred to methylcellulose membranes. Antibodies against β-actin (1:10,000; Sigma, Cat# A1978), VHL (1:1000; Cell Signaling, Danvers, MA, Cat# 2738), HIF1α (1:200; Santa Cruz, Cat# sc-53546), HIF2α (1:500; Novus, Cat# NB100-122), or CPT1A (1:200; described[38] were used to decorate the membranes, and developed by enhanced chemiluminescence. Uncropped images are provided as supplementary data (Supplementary Fig. 1).

**Chromatin immunoprecipitation (ChIP)**. ChIP was performed as previously described[39]. Briefly, semiconfluent 15 cm plates were fixed with 1% formaldehyde for 10 min and quenched with glycine. Samples were then scraped and pelleted, and nuclei were prepared with SDS lysis under hypotonic conditions. Sonication was used to reduce genomic DNA size to the range of 500–1000 base pairs. The samples were quantified, and 1 mg of chromatin was used to immunoprecipitate with 5–10 µg of indicated antibodies. Primer sequences were neg: 5′-CCCAT-GAAGAAGCTCAGGTC-3′, 5′-AGGCCATTTGAAGTGGTCAG-3′; 1: 5′-ATTA-TAGGCACCTGCCACCA-3′, 5′-CTAGCACTTCGGGGCTGAG-3′; 2: 5′-ACCCAGCTGGTCCTCTTTTA-3′, 5′-ACAAACATGAGCCACTGCAC-3′; 3: 5′-CAAAAATCAGCCGTGGTG-3′, 5′-TCTCTCTTCGTCACCCAGTC-3′; 4: 5′-ATCCCGTGTTCAGAGCAGAC-3′, 5′-AGGACAGAACAGGGTGATG-3′; 5: 5′-CATGGTGATGCACGTCTGTA-3′, 5′-CCGTTATGTCCACAAATTCCTT-3′; 6: 5′-CGGTGAGAATGACAGTCCAG-3′, 5′-TTGTCTGGACCACCAGTGAG-3′; 7: 5′-CCTGTGTCACCAAAATGTCG-3′, 5′-CCCCACCATTGCAGGTATAA-3′; 8: 5′-GCTTCTTTCGGTTTGTCAGG-3′, 5′-ATCGGCCCTCATCTTTGAGT-3′; CPT1A proximal to start site: 5′-CTCAGCCAATCCGCTGCT-3′, 5′-CCTCACC-GAGTCAGCTACG-3′; GLUT1: 5′-CTAGGGGAGCAGACGGAGAG-3′, 5′-GAGCACATGGCCTCCTTCC-3′.

**Fatty acid uptake assay**. The QBT™ fatty acid uptake assay kit was used (Molecular Devices). Briefly, BODIPY-dodecanoic acid fluorescent fatty acid analog (BODIPY 500/512, C1, C12) was added to the medium of 80,000 cells seeded in triplicate in 24-well plates, and the cells were observed over a 2-day period. Fluorescence was measured at indicated time intervals on a Tecan GENios fluorescent plate reader.

**DNA constructs**. Lentiviral transduction of shRNA was performed with pLKO.1 plasmids from Sigma: HIF1α (TRCN0000003809), HIF2α (TRCN0000003803), CPT1A (TRCN000036279, TRCN000036282, TRCN000036283), Snail

(TRCN000063819), and DEC1 (TRCN0000013249 and TRCN0000013251). shGFP was used as a control. CPT1AM overexpression was accomplished by adenoviral transfection as described[40]. Cells were infected at a multiplicity of infection of 20. Doxycycline-inducible CPT1AM expression was performed by cloning CPT1AM into pTRE2 and cotransfecting with pMA2640 into 786-O cells. Clonal lines were created.

**Mouse tumor assays**. Tumor assays were performed in 8-week-old athymic female BALB/c nude mice obtained from an in-house colony. Subcutaneous injections were performed with $2 \times 10^6$ 786-O cells in 100 µl of saline. Tumor growth was monitored and measured twice weekly with calipers; tumor volumes were calculated by the formula $V = 1/2 \, L \times W^2$. Tumors were removed and portions were prepared for immunoblot analysis, and portions were rapidly frozen in OCT on dry ice. Eight micron sections were cut and fixed in 3.7% paraformaldehyde for 30 min, and followed by blocking (1% BSA in PBS) for 30 min, 1:5000 anti-CPT1A primary Ab, and 1:5000 secondary Ab (anti-rabbit-conjugated Alexa Fluor- 488 nm, Invitrogen). Serial sections were stained with Oil Red O as described above. For doxycycline induction experiments, animals were fed Dox Diet (S3888) or control diet (S4207) from Bioserv.

**CPT1A activity measurements**. Mitochondria from normal and tumor tissue were isolated after tissue homogenization in an isotonic cold sucrose solution (30% sucrose, 1x PBS) and separated by centrifugation for 15 min at 600x$g$, and the supernatant was centrifuged again for 20 min at 12,000x$g$. The pellet was re-suspended in 2 ml of homogenization buffer (70 mM sucrose, 220 mM manitol, 5mM HEPES, 2 mM EDTA pH 7.4), centrifuged for 10 min at 7000x$g$, washed and finally re-suspended in 0.5 ml of homogenization buffer. The concentration of the mitochondria was quantified by BCA assay; 10 µg was used for the CPT1 activity assay. The malonyl-CoA-sensitive carnitine palmitoyltransferase activitiy was measured as described[41] except that the production of palmitoylcarnitine was measured using HPLC-MS/MS[42].

**Metabolic readouts**. Oxygen consumption was measured on a Seahorse XFp Analyzer according to the manufacture's recommendations. Citrate and acetyl-CoA were measured using gas chromatography mass spectrometry as described[43–45]. Acetyl-CoA was assayed at the University of Pennsylvania Quantitative Proteomics Core.

**VHL sequencing**. To determine VHL status on patient samples, genomic DNA was prepared and exons 1, 2, and 3 were PCR amplified and sequenced. Mutations and deletions were revealed by comparison to NCBI sequences.

**Statistics**. All assays were performed at least three times, and results are expressed as means ± standard deviations. Analyses were performed with GraphPad Prism 6.01. Unpaired two-tailed Student's $t$ tests or two-way ANOVA were performed to determine significance. $p$ values ≤0.05 were considered significant.

**Study approval**. Human tissue samples were collected under IRB approval (Cleveland Clinic Foundation, IRB 4630) with written informed consent of all patients. IACUC approval was obtained for all animal experiments (Case Western Reserve University IACUC 0155).

**Data availability**. The authors declare that all data supporting the findings of this study are available within the article or from the corresponding author on reasonable request.

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

## Acknowledgements

This work was supported by the following grants: American Cancer Society 121762-RSG-12-097-01-CCG, NIH R21 CA178157-01A1, AACR 14-60-36-WELF, and CTSC 4UL1TR000439. The Cytometry and Microscopy Core Facility of the Case Comprehensive Cancer Center, which is supported by P30CA43703, was used in this study. We also thank Thomas F. Peterson, Jr, for his generosity.

## Author contributions

W.D. designed and performed the experiments and wrote the paper; L.Z., A.B.-M., and B.A. performed the experiments; J.K. designed the experiments; C.L.H. designed and performed the experiments; M.P. designed the experiments; D.S., L.H., B.I.R., and S.C. contributed reagents; S.M.W. designed the experiments and wrote the paper.

## Additional information

**Competing interests:** The authors declare no competing financial interests.

