## [Peer Review File · Nature Communications]

Reviewers' comments:

Reviewer #1:

(Remarks to the Author):

The manuscript by Du et al., reports a novel mechanism for the elevated intracellular lipid storage observed in renal cancers of the clear cell phenotype. CPT1A, one of the enzymes controlling the import of fatty acids into the mitochondrial where they undergo beta oxidation, is repressed in cell lines lacking the tumor suppressor pVHL. They go on to identify a potential mechanism for the Hif-dependent hypoxic repression of CPT1A expression implicating the epigenetic marker H3K9me3. The authors show convincingly that CPT1A mRNA and protein levels are downregulated under hypoxic conditions and when HIFs are stabilized by the genetic loss of VHL. Exogenous overexpression of CPT1A decreased lipid droplets and slowed down the growth of xenografted tumors in mice. This mechanism therefore appears to contribute to increased lipid droplet formation, but I think it is possible that many other mechanisms could contribute as well. For example, increased expression of genes that upregulate lipid uptake (CD36) or synthesis (SREBP1) could also contribute to the phenotype.

Major Comments:

- 1) The authors' model is one of increased lipid oxidation due to CPT1A expression that decreases lipid storage. However, data on CPT1A enzymatic activity in the matched +/-VHL cell lines and normoxic/hypoxic conditions are not shown. It is important to correlate rates of beta oxidation with the observed lipid storage. The CPT1 activity measured in 5 clinical specimens is helpful, however, the values are not correlated to either VHL status, CPT1A levels or lipid droplet abundance.
- 2) The postconfluence induction of lipid droplet formation requires some additional discussion and expansion, since it is not clear what triggers the phenomenon which is missing in regular growth conditions. Are the cells still dividing or have undergone contact inhibition? Are the CPT1A levels similar between low density and post-confluent cells? Are the growth media refreshed during the 8 day incubation or is this a response to a nutrient deprivation stress?
- 3) The fatty acid analog uptake rates shown in fig3 plateau after 20-30min, how do the authors interpret that? Additionally, the immunofluorescence staining in the +VHL cells appears bright but diffuse. Do the authors know if the fatty acid-BODIPY undergoes any modifications/oxidation or not? Is the decrease in fluorescence at 120 minute statistically significant? Why do the authors think this decrease occurs?
- 4) The authors report that the combination of serum and glutamine withdrawal causes significant loss of viability but that the surviving cells do accumulate lipid droplets. Are these remaining cells able to recover from the stress and divide or have they undergone permanent arrest? What fraction of cells remains on the dish after this treatment?
- 5) It is interesting that the authors are invoking a new activity of HIF1 as a transcriptional repressor. Is stabilization of HIF1a sufficient for this activity, or are additional, hypoxia-dependent activities necessary? The ChIP assays are performed in VHL-reconstituted cells under normoxia and hypoxia. It is important to clarify if the normoxic CPT1A repression in the VHL null cells occurs through the same binding sites and whether similar pathways and epigenetic modifications are found to regulate the promoter.

Reviewer #2:

(Remarks to the Author):

The authors Du and colleagues perform studies to investigate mechanism that underlie fatty acid accumulation in RCC of the clear cell type, which is a phenotypic hallmark of this tumor (in addition to glycogen accumulation). Previous studies have implicated PGC-1 α dysregulation, dysregulation of lipid binding protein expression (Adfp) and decreased FA oxidation and mitochondrial dysfunction. The exact contributions of these individual processes to increased FA

accumulation are not clear.

The authors now provide evidence from in vitro and in vivo studies, and human tumor samples that carnitine palmitoyltransferase 1A (CPT1A) is suppressed in a HIF-dependent manner in VHL-/- RCC cells. Forced expression of CPT1A suppresses the lipid phenotype in VHL-/- cells and more importantly suppressed tumor growth in a xenograft model. Chip sequencing identified HIF binding sites in the CPT1A locus.

Major critique: The authors address an important question in RCC biology. While strong experimental and clinical evidence is provided that CPT1A is suppressed in RCC (in contrast to other tumors), the reviewer has several questions regarding the biological role of CPT1A suppression and how its role in RCC FA metabolism differentiates itself from other observations that were made with regard to lipid dysregulation in this context. Also, some additional studies should be included concerning the metabolic characterization of engineered cell lines. The role of FA accumulation in tumor growth is mechanistically unclear, and this should be discussed in more detail.

1. To what degree are very long chain FA affected. In theory they should not be. Is this the case here.
2. What are the intracellular levels of citrate and acetyl CoA after manipulating CPT1A expression.
3. Can the authors include functional measurements of oxygen consumption rates (OCR) after manipulating (increasing and decreasing expression of) CPT1A. OCR should follow CPT1 expression levels. The more CPT1A expression the higher the OCRs.
4. How do these data relate to recent studies concerning PGC1- α suppression and the role of perilipin.
5. It is unlikely that HIF functions as direct transcriptional repressor. There is currently not sufficient experimental evidence to support a suppressive role for HIF. The authors would need to investigate this in more detail using standard transcriptional assays.

Minor critiques:

Page 9: "Thus CPT1A repression by HIF pathway is capable of reversing the effects of VHL deficiency regarding lipid storage". Shouldn't this be VHL re-constitution.
To verify that lipid accumulation was dependent on

Page 10/11: "CPT1A expression in this subclone, cells were first allowed to form droplets in vitro in the presence of doxycycline or vehicle (dimethylsulfoxide (DMSO))". Please clarify this.

Reviewer 1:

it is possible that many other mechanisms could contribute as well...

We certainly agree that several mechanisms likely contribute to the lipid deposition phenotype, but we have focused on what we think is a bottleneck point, and a novel effector in RCC. We have tested directly the role of CD36 which we expected to participate in the phenotype but failed to find a contribution at least in RCC4 cells (shown in Figure 3D), unlike other cells and tissues referenced in the manuscript. We do not however rule out other mechanisms of lipid uptake contributing.

1) The authors' model is one of increased lipid oxidation due to CPT1A expression that decreases lipid storage. However, data on CPT1A enzymatic activity in the matched +/-VHL cell lines and normoxic/hypoxic conditions are not shown. It is important to correlate rates of beta oxidation with the observed lipid storage. The CPT1 activity measured in 5 clinical specimens is helpful, however, the values are not correlated to either VHL status, CPT1A levels or lipid droplet abundance.

CPT1A enzymatic activity is now included for both +/- VHL cell lines and for CPT1A overexpression cells in Figures 4D and 6C, respectively. We have also obtained new patient samples and measured CPT1A activity, CPT1A mRNA, determined VHL status, and stained for ORO. The data are presented in Figure 8. We were unable to correlate lipid droplet abundance to activity directly because the single slices used for ORO staining are not necessarily representative of non-homogeneous tumor chunks needed to produce mitochondrial for activity assays. Thus the most useful predictor of CPT1A for tumor behavior appears to be the message expression level, as shown in Figure 8B.

2) The postconfluence induction of lipid droplet formation requires some additional discussion and expansion, since it is not clear what triggers the phenomenon which is missing in regular growth conditions. Are the cells still dividing or have undergone contact inhibition? Are the CPT1A levels similar between low density and post-confluent cells? Are the growth media refreshed during the 8 day incubation or is this a response to a nutrient deprivation stress?

We have addressed this comment by testing proliferation in post confluent cells by BrdU labeling, and found that the cells indeed remain proliferative (Figure 2F). We measured CPT1A levels in low and high density cell conditions and found that there is an additional reduction (~25%) at post confluent levels that may help explain the apparent increased deposition (Figure 4H). Finally, we tested changing growth media during the deposition time and found no effect, arguing against nutrient deprivation (Figure 2E).

3) The fatty acid analog uptake rates shown in fig3 plateau after 20-30min, how do the authors interpret that? Additionally, the immunofluorescence staining in the +VHL cells appears bright but diffuse. Do the authors know if the fatty acid-BODIPY undergoes any modifications/oxidation or not? Is the decrease in fluorescence at 120 minute statistically significant? Why do the authors think this decrease occurs?

We took direction for this assay from published papers on 3T3L1 adipocytes which frequently are either done over periods of seconds for short term uptake, or over ~60 minutes for long term uptake where similar plateau behavior is found (Dubikovskaya, et al, Methods Enzymology 2014; 538:107-134, PMID: PMC4269161), and from Wang, et al (Mol Biol Cell 2010; 21:1991-2000, PMID: PMC2883943) where in

hepatocytes the metabolism of the fluorescent tracer is demonstrated by incorporation of the Bodipy FA into glycerolipids and esterification products. Presumably a competition of uptake and metabolism/turnover describes the function of plateau here. Further, we thus hypothesize that the differing localization of signal in VHL + and – cells is indicative of the differing utilization of the labeled lipid. We agree with the reviewer that the drop in signal at 120 minutes is unexpected, and thus have removed the time point as our interest is in observing any changes in uptake per se. As mentioned, we also investigated the role of CD36 in uptake in figure 3.

4) The authors report that the combination of serum and glutamine withdrawal causes significant loss of viability but that the surviving cells do accumulate lipid droplets. Are these remaining cells able to recover from the stress and divide or have they undergone permanent arrest? What fraction of cells remains on the dish after this treatment?

The loss of viability of cells with glutamine withdrawal was a starting point for us, as recently published by Gameiro, et al., 2013. Based on the reviewer's comment, we tested the viability of remaining cells by rescuing them with fresh media at various time points and performing a clonogenic assay. We found that survival was dramatically dependent on time of glutamine and serum deprivation, and that indeed remaining cells could come back when rescued. These data are incorporated into figure 2.

5) It is interesting that the authors are invoking a new activity of HIF1 as a transcriptional repressor. Is stabilization of HIF1a sufficient for this activity, or are additional, hypoxia-dependent activities necessary? The ChIP assays are performed in VHL-reconstituted cells under normoxia and hypoxia. It is important to clarify if the normoxic CPT1A repression in the VHL null cells occurs through the same binding sites and whether similar pathways and epigenetic modifications are found to regulate the promoter.

The role of HIF as a repressor is unusual, but not without precedent, as reviewed recently by Cormac Taylor (Cavadas, et al, Exp Cell Res, 2017). Based on the reviewer comment, we dug a bit deeper by testing two common repressor proteins that are known to be HIF activated targets: DEC1 and Snail. Inhibition of neither DEC1 nor Snail however rescued CPT1A expression, while controls for each were affected. This does not rule out all repressors, and leaves the door open for others we have not yet tested. Regarding the normoxic/hypoxic question, we have observed the effects on CPT1A in normoxia with VHL loss, as well as with RCC4VHL cells treated with DMOG in normoxia, telling us that other hypoxic effects, such as epigenetic modifications with oxoglutarate dependent dioxygenases like the Jumanji proteins are likely not directly involved. New data are included in figure 5.

Reviewer #2:

The role of FA accumulation in tumor growth is mechanistically unclear, and this should be discussed in more detail.

Based on comments from reviewer 1, we added additional studies investigating rescuing of glutamine and serum starved cells (Figure 2) that contained lipid deposits that strengthen the argument that the droplets provide a survival advantage as has been suggested both by Adrian Harris and Celeste Simon. We have

added to the discussion to this point as well.

1. To what degree are very long chain FA affected. In theory they should not be. Is this the case here.

In our newly added CoA analyses (Figure 6) we assessed up to 18-carbon fatty acids, but could not detect sufficient concentrations of the longer chain species with our existing protocol. We agree with the reviewer that we would not expect to see changes in VLCFA due to CPT1A alteration, particularly as the VLCFA are such a small fraction of the cellular fatty acids and are metabolized in the peroxisomes. Since we were focused on mitochondrial metabolism, we felt that this was beyond our scope.

2. What are the intracellular levels of citrate and acetyl CoA after manipulating CPT1A expression.

Citrate and AcCoA measurements are added to figure 6. As expected, CPT1A expression caused an increase in citrate, but interestingly AcCoA levels did not significantly change perhaps due to a compensatory change.

3. Can the authors include functional measurements of oxygen consumption rates (OCR) after manipulating (increasing and decreasing expression of) CPT1A. OCR should follow CPT1 expression levels. The more CPT1A expression the higher the OCRs

OCR measurements were added to Figure 6. Indeed, as the reviewer noted, OCR follow CPT1 expression.

4. How do these data relate to recent studies concerning PGC1- α suppression and the role of perilipin.

The roles of PGC1 α and perilipin were addressed by determining if there is an interaction of CPT1A regulation and either PGC1 α or PLIN2 in Figure 5. First we assessed whether DEC1 suppression, which causes elevation of PGC1 α has any effect on CPT1A (which it does not). Then we determined if CPT1A suppression by shRNA affects PGC1 α , which it does. Thus the data suggest some overlap between CPT1A and PGC1 α at the expression level, but the HIF dependent mechanism of PGC1 α suppression is unrelated to CPT1A. PLIN2 was unaffected in similar assays.

5. It is unlikely that HIF functions as direct transcriptional repressor. There is currently not sufficient experimental evidence to support a suppressive role for HIF. The authors would need to investigate this in more detail using standard transcriptional assays.

We fully agree with this comment. Describing HIF as a suppressor requires more in-depth transcriptional assays. We have added analyses of two popular HIF activated transcriptional repressors (DEC1 and Snail) but did not find a link to CPT1A. While we continue to investigate the exact mechanism, we have relaxed our assertion that HIF is a repressor in this case, and have added to the discussion on this regard.

Minor critiques:

Page 9: "Thus CPT1A repression by HIF pathway is capable of reversing the effects of VHL deficiency regarding lipid storage". Shouldn't this be VHL re-constitution.

To verify that lipid accumulation was dependent on

Yes, thank you for the comment on a very confusing sentence. This has been changed to “Thus CPT1A repression by the HIF pathway mediates the effects of VHL loss regarding lipid storage.”

10/11: “CPT1A expression in this subclone, cells were first allowed to form droplets in vitro in the presence of doxycycline or vehicle (dimethylsulfoxide (DMSO))”. Please clarify this.

Again, thanks for pointing this out. Sentence changed to “To verify that lipid accumulation was affected by CPT1A expression in this subclone, cells were first tested for droplet formation in vitro in the presence of doxycycline or vehicle (dimethylsulfoxide (DMSO)).”

Reviewers' comments:

Reviewer #1 (Remarks to the Author):

The authors have added significant data to the manuscript and improved the overall quality of the science. However, the interpretation of Hif1a as a transcriptional repressor is still weak. That addition of the Dec1 and Snail data is something of a red herring, as the authors are claiming direct HIF binding to the CPT1a promoter, and Dec1 and Snail would act indirectly with alternative binding sites. The authors state in their response letter that "we have observed the effects on CPT1A in normoxia with VHL loss" but do not clarify if this binding is at the CPT1a promoter or just Cpt1a mRNA levels. Furthermore, they still do not show the normoxic HIF1a chromatin binding data from VHL negative cells in figure 5.

They mention/show DMOG data, which appears to better at generating HIF1a chromatin binding (60 fold vs 20 fold for hypoxia) but less Cpt1a mRNA repression (maybe 50% versus 80% for hypoxia figs 5a and 5c). Do the authors interpret this to mean Hif1a is acting without additional co-repressor binding? However, since DMOG is an aKG mimetic, this could interfere with Jumanji demethylases affecting H3K9 me3 (which could be important as pointed out in the response to question 5). It appears that additional clarification in this mechanism would increase the impact of the conclusions.

Reviewer #2 (Remarks to the Author):

The authors have answered my critique to my complete satisfaction. I have no additional comments.

Reviewer #1 (Remarks to the Author):

The authors have added significant data to the manuscript and improved the overall quality of the science. However, the interpretation of Hif1a as a transcriptional repressor is still weak. That addition of the Dec1 and Snail data is something of a red herring, as the authors are claiming direct HIF binding to the CPT1a promoter, and Dec1 and Snail would act indirectly with alternative binding sites. The authors state in their response letter that “we have observed the effects on CPT1A in normoxia with VHL loss” but do not clarify if this binding at the CPT1a promoter or just Cpt1a mRNA levels. Furthermore, they still do not show the normoxic HIF1a chromatin binding data from VHL negative cells in figure 5.

The reviewer appropriately points out that the presentation of Dec1 and Snail data is unclear and confusing from the perspective of the mechanism that we have observed (HIF direct suppression of *CPT1A*). This was an oversight due to the edits from the first round, for which we added data that in retrospect were presented in a non-logical flow. We have now amended the paper to present the data more logically, and hope the data no longer stand out as odd. We have also added to the discussion to clarify the “effects on CPT1A” in normoxia with VHL loss relate to gene expression. This is particularly relevant in the context of the response to the second comment. We also now show the binding of HIF1a in VHL negative cells in Figure 5I.

They mention/show DMOG data, which appears to better at generating HIF1a chromatin binding (60 fold vs 20 fold for hypoxia) but less Cpt1a mRNA repression (maybe 50% versus 80% for hypoxia figs 5a and 5c). Do the authors interpret this to mean Hif1a is acting without additional co-repressor binding? However, since DMOG is an aKG mimetic, this could interfere with Jumanji demethylases affecting H3K9 me3 (which could be important as pointed out in the response to question 5). It appears that additional clarification in this mechanism would increase the impact of the conclusions.

The interpretation of HIF functioning with or without co-repressor binding is simply too soon to tell, and quite honestly untested here. We do find that the effects of *CPT1A* suppression occur in normoxia with VHL loss, with DMOG treatment, and with hypoxia, leading us to suggest that the induction of hypoxia-dependent co-repressors whose activity depends on low oxygen is unnecessary for the down regulation of expression. While the reduction in expression in RCC4VHL with DMOG is less than with hypoxia (from panel A and H) as the reviewer notes, the reduction without VHL (panel A) is just as impressive as hypoxia. Thus we would not want to over interpret the “hypoxia” effect from the DMOG effect. We added to the discussion to clarify this fact, as well as to suggest where future work is needed.

REVIEWERS' COMMENTS:

Reviewer #1 (Remarks to the Author):

The authros have addressed all comments.